# Mediterranean Diet and Healthy Eating in Subjects with Prediabetes from the Mollerussa Prospective Observational Cohort Study

**DOI:** 10.3390/nu13010252

**Published:** 2021-01-16

**Authors:** Mireia Falguera, Esmeralda Castelblanco, Marina Idalia Rojo-López, Maria Belén Vilanova, Jordi Real, Nuria Alcubierre, Neus Miró, Àngels Molló, Manel Mata-Cases, Josep Franch-Nadal, Minerva Granado-Casas, Didac Mauricio

**Affiliations:** 1Primary Health Care Centre Cervera, Gerència d’Atenció Primaria, Institut Català de la Salut, 25200 Lleida, Spain; mireiafalguera@hotmail.com; 2Department of Medicine, Lleida Institute for Biomedical Research Dr. Pifarré Foundation IRB Lleida, University of Lleida, 25198 Lleida, Spain; mbvilanova@gmail.com (M.B.V.); nurialcubierre@gmail.com (N.A.); 3Department of Endocrinology & Nutrition, Hospital de la Santa Creu i Sant Pau & Institut d’Investigació Biomédica Sant Pau (IIB Sant Pau), 08041 Barcelona, Spain; esmeraldacas@gmail.com (E.C.); nut.marina.rojo.l@gmail.com (M.I.R.-L.); 4Center for Biomedical Research on Diabetes and Associated Metabolic Diseases (CIBERDEM), Instituto de Salud Carlos III, 08907 Barcelona, Spain; jordi.real@gmail.com (J.R.); manelmatacases@gmail.com (M.M.-C.); josep.franch@gmail.com (J.F.-N.); 5Primary Health Care Centre Igualada Nord, Consorci Sanitari de l’Anoia, Institut Català de la Salut, 08700 Barcelona, Spain; 6DAP-Cat Group, Unitat de Suport a la Recerca Barcelona, Fundació Institut Universitari per a la Recerca a l’Atenció Primària de Salut Jordi Gol i Gurina (IDIAPJGol), 08007 Barcelona, Spain; 7Primary Health Care Centre Tàrrega, Gerència d’Atenció Primaria, Institut Català de la Salut, 25300 Lleida, Spain; miro.vallve@gmail.com; 8Primary Health Care Centre Guissona, Gerència d’Atenció Primaria, Institut Català de la Salut, 25210 Lleida, Spain; angelsmollo@gmail.com; 9Primary Health Care Centre Raval Sud, Gerència d’Atenció Primaria Barcelona, Institut Català de la Salut, 08001 Barcelona, Spain; 10Lleida Institute for Biomedical Research Dr. Pifarré Foundation IRBLleida, University of Lleida, 25198 Lleida, Spain; 11Faculty of Medicine, University of Vic (UVIC/UCC), 08500 Vic, Spain

**Keywords:** prediabetes, Mediterranean diet, healthy eating, dietary pattern, nutrient intake, physical activity

## Abstract

We aimed to assess differences in dietary patterns (i.e., Mediterranean diet and healthy eating indexes) between participants with prediabetes and those with normal glucose tolerance. Secondarily, we analyzed factors related to prediabetes and dietary patterns. This was a cross-sectional study design. From a sample of 594 participants recruited in the Mollerussa study cohort, a total of 535 participants (216 with prediabetes and 319 with normal glucose tolerance) were included. The alternate Mediterranean Diet score (aMED) and the alternate Healthy Eating Index (aHEI) were calculated. Bivariable and multivariable analyses were performed. There was no difference in the mean aMED and aHEI scores between groups (3.2 (1.8) in the normoglycemic group and 3.4 (1.8) in the prediabetes group, *p* = 0.164 for the aMED and 38.6 (7.3) in the normoglycemic group and 38.7 (6.7) in the prediabetes group, *p* = 0.877 for the aHEI, respectively). Nevertheless, women had a higher mean of aMED and aHEI scores in the prediabetes group (3.7 (1.9), *p* = 0.001 and 40.5 (6.9), *p* < 0.001, respectively); moreover, they had a higher mean of aHEI in the group with normoglycemia (39.8 (6.6); *p* = 0.001). No differences were observed in daily food intake between both study groups; consistent with this finding, we did not find major differences in nutrient intake between groups. In the multivariable analyses, the aMED and aHEI were not associated with prediabetes (odds ratio (OR): 1.19, 95% confidence interval (CI): 0.75–1.87; *p* = 0.460 and OR: 1.32, 95% CI: 0.83–2.10; *p* = 0.246, respectively); however, age (OR: 1.04, 95% CI: 1.02–1.05; *p* < 0.001), dyslipidemia (OR: 2.02, 95% CI: 1.27–3.22; *p* = 0.003) and body mass index (BMI) (OR: 1.09, 95% CI: 1.05–1.14; *p* < 0.001) were positively associated with prediabetes. Physical activity was associated with a lower frequency of prediabetes (OR: 0.48, 95% CI: 0.31–0.72; *p* = 0.001). In conclusion, subjects with prediabetes did not show a different dietary pattern compared with a normal glucose tolerance group. However, further research is needed on this issue.

## 1. Introduction

Prediabetes is a condition defined as an impaired fasting glucose (IFG) and/or isolated impaired glucose tolerance (IGT) with a high risk of developing type 2 diabetes (T2D) [1]. According to the Di@bet.es study, the prevalence of prediabetes was approximately 14.8% in the Spanish adult population [2]. The implementation of therapeutic strategies focused on changes in lifestyle, such as physical activity and a healthy dietary pattern (i.e., Mediterranean diet (MedDiet)), could reduce the risk of developing T2D in subjects with prediabetes [3,4]. The dietary patterns (MedDiet and healthy eating) have been assessed using a variety of dietary quality indexes [5]. However, the alternate Mediterranean Diet score (aMED) and the alternate Healthy Eating Index (aHEI) have demonstrated a potential association with the risk of developing T2D and cardiovascular diseases [6,7]. Therefore, these two dietary quality indexes are able to assess the adherence to the MedDiet and healthy eating in a population with prediabetes.

Several studies have evaluated the potential relationship between dietary patterns and prediabetes, albeit with some conflicting results [8,9,10,11,12,13,14,15,16,17,18,19,20,21,22,23,24,25]. Two randomized clinical trials performed in subjects with prediabetes demonstrated that modifications in lifestyle (e.g., physical activity, nutrition and lifestyle counseling) improved fasting glucose and insulin resistance, thereby reducing the risk of developing T2D [8,9]. A prospective study observed that a healthier dietary pattern was associated with a lower risk of prediabetes in men [10]. Moreover, a higher intake of whole grain was associated with a decreased risk of prediabetes in healthy subjects [11]. Another prospective study found that a Western dietary pattern could be a risk factor for the development of IGT [12]; in addition, other researchers observed that a healthy dietary pattern could prevent hyperinsulinemia in healthy subjects. Regarding the MedDiet, a cross-sectional study performed using subjects without diabetes observed an inverse relationship between the MedDiet and IFG [24]. Additionally, a cross-sectional study from the Di@bet.es study performed in a group of subjects with prediabetes and unknown diabetes found that a higher adherence to the MedDiet was associated with a lower prevalence of these conditions [25]. On the other hand, three prospective studies performed with large samples did not find any association between dietary patterns such as the MedDiet and the risk of prediabetes [13,14,15]. A case-control study observed a positive association between a dietary pattern based on sweets, fats, meat and mayonnaise and prediabetes [16], while a dietary pattern based on vegetables, fruits and legumes was related to a lower risk of prediabetes [16,17]. Cross-sectional studies observed that a diet rich in vegetables, fruits, whole grains, fiber, legumes and dairy products was associated with a low risk of prediabetes [18,19,20,21]. However, a recent meta-analysis of observational studies found that a dietary pattern rich in carbohydrates, fat, animal protein and omega-3 was associated with an increased risk of prediabetes [22].

To our knowledge, this is the first study performed to assess the potential relationship between dietary pattern (i.e., MedDiet and healthy eating) and the prevalence of prediabetes in a nonurban population using the aMED and aHEI scores. We hypothesized that a higher adherence to the MedDiet and healthy eating patterns would be associated with a lower prevalence of prediabetes. Thus, the aim of the study was to assess the relationship between dietary patterns and the prevalence of prediabetes in a semirural population. Secondarily, we aimed to investigate factors associated with prediabetes.

## 2. Materials and Methods

### 2.1. Study Design

This was a cross-sectional study from the Mollerussa prospective cohort study in Catalonia, a semirural area in this region. The primary objective of this study was to assess the prevalence of prediabetes. This population-based cohort has been described in detail in a previous publication [26]. From a total sample of 594 participants, 20 participants with undiagnosed diabetes were excluded; a further 39 subjects were excluded due to a lack of glycemic and nutritional data. A final sample of 535 subjects was included: 216 individuals with prediabetes and 319 individuals with a normal glucose tolerance. The inclusion criteria were subjects aged over 25 years attending a primary health care center in the area. The exclusion criteria were as follows: diagnosis of any type of diabetes mellitus, treatment with antidiabetic drugs, systemic glucocorticoids or beta blockers, the presence of cardiovascular disease, cancer, kidney disease, anemia, hepatitis, gastrointestinal diseases, recent abdominal surgery, chronic infectious diseases, chronic obstructive pulmonary disease or major psychiatric disorders, as well as the presence of any condition requiring any diet treatment [26,27]. Prediabetes was determined in subjects with fasting plasma glucose from 100 mg/dL to <126 mg/dL or glycated hemoglobin (HbA1c) from 5.7% to <6.5%. Normal glucose tolerance was defined as a fasting plasma glucose (FPG) up to <100 mg/dL and HbA1c up to <5.7%. Hypertension and dyslipidemia were considered if individuals were using specific agents to treat these two conditions.

### 2.2. Assessment of the Dietary Pattern

We used a semiquantitative food frequency questionnaire (FFQ) validated in subjects from Spain. The FFQ was administered to every study participant by a trained researcher [28]. The questionnaire contained 101 items that collect data on usual food consumption over a period of one year prior to the study visit. The dietary pattern was assessed using 2 different scores: the alternate Mediterranean Diet score (aMED) and the alternate Healthy Eating Index (aHEI). The aMED score included monounsaturated-to-saturated fat ratio, legumes, vegetables, nuts, fruits, nuts, cereals, fish, meat and wine. This score ranged from 0 to 9, with higher scores indicative of a higher adherence to the MedDiet [29]. The aHEI included fruit, nuts, vegetables, soy, red and white meats, cereal fiber, polyunsaturated-to-saturated fatty acid ratio, trans fat and alcohol consumption [7]. This score ranged from 2.5 (unhealthy diet) to 87.5 (healthy diet), excluding long-term multivitamin use. Food consumption and nutrient intake data were obtained from the US Department of Agriculture food composition tables, as well as other sources for Spanish and English foods and portion sizes [30,31,32]. Furthermore, the Spanish food composition tables were used to prevent an overestimation of nutrient intake from fortified dairy products in the US. Finally, the nutrient intake was adjusted for energy intake.

### 2.3. Clinical Data

Medical records of participants were thoroughly reviewed to collect clinical variables. A standardized questionnaire was individually administered at baseline. Blood samples were gathered from each of the study participants. Anthropometric measures, e.g., body weight, height and waist, were determined using standard methods as previously described [26,27]. We categorized the educational level as follows: (a) no university, for those participants without a university degree; (b) graduate or higher if participants had a university education. Physical activity was evaluated using the Spanish-validated International Physical Activity Questionnaire [33]; this was classified as sedentary or active (not regularly or regularly active, respectively). Tobacco exposure was defined as current and former smokers. All study participants signed an informed consent form according to the Declaration of Helsinki principles. The study was approved by the Ethics Committee of the Primary Health Care University Research Institute (IDIAP) Jordi Gol (P12/043).

### 2.4. Statistical Analysis

Descriptive statistics were performed using mean and standard deviation (SD) for continuous variables, and the absolute (number) and relative (percentage) frequencies for categorical variables. The aMED and aHEI were stratified in tertiles, from T1 (poorer adherence) to T3 (better adherence). Statistical significance was calculated using the Chi-squared test to determine the differences for categorical variables. The Student’s *t*-test was used to analyze the differences between quantitative variables. Adjustment methods included the Bonferroni correction, in which the *p*-values are multiplied by the number of comparisons. Less conservative corrections were also included by Holm (1979), Hochberg (1988), Hommel (1988), Benjamini and Hochberg (1995) (BH) or false discovery rate (fdr) and Benjamini and Yekutieli (2001) (BY), respectively [32]. A pass-through option (none) was also included. Multivariable logistic models were performed adjusting for potential confounders, defined as variables statistically significant in the bivariate analysis or clinically associated with diabetes, using the Enter method to support the hypothesis of the study. We assessed the goodness-of-fit assumption using the Hosmer–Lemeshow test for logistic models. To estimate the measures, we used the odds ratio (OR) with 95% confidence interval (95% CI). A *p*-value < 0.05 was considered statistically significant. The analysis was performed using R statistical software [34].

## 3. Results

The clinical characteristics of the study groups are shown in Table 1. Subjects with prediabetes were older (mean (SD) = 54.8 (12.2) and 47.5 (12.8); *p* < 0.001, respectively) and had a lower educational level (94.4% and 85.1%; *p* = 0.001, respectively), higher body mass index (BMI) (mean (SD) = 27.5 (4.8) and 25.3 (4.3); *p* < 0.001, respectively), waist (mean (SD) = 97.0 (12.4) and 92.0 (11.9); *p* < 0.001, respectively), hypertension (22.2% and 13.2%; *p* = 0.009, respectively), dyslipidemia (32.9% and 16.0%; *p* < 0.001, respectively), HbA1c (mean (SD) = 5.8 (0.3) and 5.3 (0.3); *p* < 0.001, respectively) and total cholesterol (TC) (mean (SD) = 205.0 (33.1) and 198.0 (38.1); *p* = 0.021, respectively) compared with the normoglycemic group. In addition, they were more sedentary (62.0% and 76.2% with regular physical activity; *p* = 0.001, respectively).

### 3.1. Dietary Pattern and Prediabetes

The dietary quality indexes of the study groups are shown in Table 2. In the group with prediabetes, the mean aMED and aHEI scores were higher in women compared with men (3.7 (1.9) and 2.9 (1.5), *p* = 0.001 in the aMED, and 40.5 (6.9) and 36.1 (6.2) in the aHEI, *p* < 0.001, respectively); moreover, a higher frequency of women in higher tertiles of aMED and aHEI was observed in comparison with men (35.2% and 18.2%; *p* = 0.018 for the aMED, and 39.1% and 15.9%; *p* < 0.001 for the aHEI, respectively). In addition, women with normal glucose tolerance showed a higher aHEI compared with men (39.8 (6.6) and 37.1 (7.9); *p* = 0.001, respectively). However, no differences were observed in the aMED and aHEI between both study groups. Regarding aMED, the mean (SD) was 3.2 (1.8) in the group with normal glucose tolerance and 3.4 (1.8) in the group with prediabetes (*p* = 0.164). For aHEI, the mean (SD) was 38.6 (7.3) in the normoglycemic group and 38.7 (6.7) in the prediabetes group (*p* = 0.877). In terms of daily food consumption, no differences were observed between both groups in any food or nutrient included in the aMED and aHEI (Appendix A). However, participants with normal glucose tolerance had a higher consumption of stearic acid in comparison with individuals with prediabetes (6.4 g/day and 5.9 g/day; *p* = 0.011, respectively) (Appendix A).

### 3.2. Clinical Factors and Prediabetes

The adjusted multivariable analysis for the aMED and aHEI scores revealed that increasing age (OR: 1.04, 95% CI: 1.02–1.05; *p* < 0.001 and OR: 1.04, 95% CI: 1.02–1.06; *p* < 0.001, respectively), the presence of dyslipidemia (OR: 1.89, 95% CI: 1.19–3.00; *p* = 0.007 and OR: 2.02, 95% CI: 1.27–3.22; *p* = 0.003, respectively) and BMI (OR: 1.09, 95% CI: 1.05–1.14; *p* < 0.001 and OR: 1.09, 95% CI: 1.05–1.14 *p* < 0.001, respectively) were associated with the presence of prediabetes (Table 3). Moreover, physical activity was associated with a low prevalence of prediabetes (OR: 0.48, 95% CI: 0.31–0.72; *p* < 0.001 for the aMED and OR: 0.49, 95% CI: 0.32–0.74; *p* = 0.001 for the aHEI) (Table 3). We could not find an interaction between sex and aHEI or sex and aMED (no statistical significance in the multivariable models; data not shown).

## 4. Discussion

Participants with prediabetes did not have a different dietary pattern in comparison with a group with normal glucose tolerance; however, women had a healthier eating pattern in comparison with men in both study groups. Regarding specific nutrient intake, subjects with normoglycemia had a higher intake of stearic acid. The aMED and aHEI were not associated with the prevalence of prediabetes. However, increasing age, dyslipidemia and BMI were associated with prediabetes. Additionally, increased physical activity was associated with a lower frequency of prediabetes.

In our study, individuals with prediabetes did not show a different dietary pattern (i.e., MedDiet and healthy eating) in comparison with a group with normoglycemia. Furthermore, the aMED and aHEI were not associated with the presence of prediabetes. These results are similar to findings of the Hoorn study, a six-year prospective study that did not find a significant association between healthy eating patterns and the incidence of prediabetes [13]. However, these authors found that a higher adherence to this pattern was associated with a lower T2D incidence. Additionally, the Coronary Artery Risk Development in Young Adults (CARDIA) study did not find any association between a modified MedDiet pattern and the risk of prediabetes in young adulthood [15]. A cross-sectional analysis from the Di@bet.es study in Spain showed that individuals with prediabetes and unknown diabetes had a lower adherence to the MedDiet in comparison with those with normal glucose tolerance [25]. However, a qualitative FFQ was used to assess adherence to the MedDiet, and the amount of nutrients per day or total energy intake could not be calculated. A randomized clinical trial performed to assess the effect of dietary intervention in subjects with prediabetes demonstrated improved post-intervention fasting glucose and insulin resistance [8]. This was similar with another clinical trial which found a reduced prevalence of prediabetes in obese subjects after an intervention based on physical activity and dietary changes [9]. Furthermore, a three-year prospective study found that a healthy dietary pattern was associated with a reduced risk of hyperinsulinemia in healthy individuals [12]. The Rotterdam study, a prospective study, found a lower prediabetes risk with a plant-based diet after adjusting for sociodemographic and lifestyle factors [14]; however, the authors did not find any association after additional adjustment for BMI. This was similar to the study by Ericson et al. that found a lower prevalence of prediabetes was associated with a higher adherence to a healthy eating pattern in adults without prediabetes [21]. The ATTICA study, a prospective large cohort study, found that a greater quality of diet was associated with a lower risk of prediabetes in men, but not women [10]. On the other hand, case-control and cross-sectional studies observed that a diet rich in vegetables and fruits was associated with a lower presence of prediabetes [16,17,19]. In addition, the Women’s Health study showed a relative risk reduction of 30% in diabetes incidence during a 20-year period with a higher adherence to the MedDiet [35]. The biomarkers of insulin resistance, BMI, lipoprotein metabolism and inflammation in the onset of diabetes could explain these differences [35].

As expected, our results indicated that women had healthier eating patterns in terms of MedDiet and healthy eating compared with men. A cross-sectional study in our region observed that women had a higher adherence to the MedDiet [36], and a cross-sectional study performed with subjects without diabetes observed that women aged over 65 years had a higher adherence to the MedDiet than men [24]. Moreover, Filippatos et al. observed that young women had a higher adherence to the MedDiet [23]. In contrast, a 10-year prospective study performed with older adults found that men had healthier eating patterns [10].

In terms of daily nutrient intake, subjects with normal glucose tolerance had a higher intake of stearic acid in comparison with the group with prediabetes. There were no differences between the food sources of this fatty acid between study groups. Furthermore, there were no differences in the intake of other fatty acids or other sources of fat. This is in contrast with a post-hoc analysis from a randomized clinical trial which found a positive association between saturated fat and impaired fasting glucose in a healthy population [18]. The European Prospective Investigations into Cancer and Nutrition (EPIC)-InterAct case-cohort study (12,132 cases with T2D and 15,164 controls from across Europe) found that even chain saturated fatty acids including stearic acid were positively correlated with the incidence of T2D [37]. However, no studies have specifically assessed the intake of stearic acid in subjects with prediabetes. In addition, stearic acid is a saturated fatty acid that generates oleic acid and elaidic acid. For these reasons, we should be cautious about this finding regarding stearic acid intake, and this should not lead to any firm conclusions.

The risk factors associated with prediabetes were increasing age, dyslipidemia and BMI. In addition, physical activity was a protective factor. These results are in line with a cross-sectional study in Korea which showed that older age, obesity and low physical activity were predictors of prediabetes [38]. Furthermore, a cross-sectional analysis of the Primary Health Care on the Evolution of Patients with Prediabetes (PREDAPS) study, a cohort study performed in primary health care in Spain, also found obesity to be a predictor of prediabetes [39]; however, physical activity was not related with this. A systematic review of randomized clinical trials reported no firm conclusions about the effectiveness of interventions with diet or physical activity alone on T2D risk, especially in subjects with high T2D risk, although the combination of the two was associated with a reduced or delayed incidence of T2D in those with IGT [40]. Overall, this suggests the potential benefits of physical activity in combination with diet and their role in the prevention of prediabetes. Clinical factors such as age, BMI and dyslipidemia play a major role in the development of prediabetes and T2D, and this should be considered further in future research.

The current study has several limitations. A causal relationship between study variables cannot be established because of the cross-sectional study design. Additionally, the sample size was smaller in comparison with similar studies; therefore, results are not comparable with other populations. Nevertheless, the study has some potential strengths. This current study is the first one that assessed the differences and relationships between two dietary patterns (MedDiet and healthy eating) and prediabetes in a semirural population. Furthermore, the aMED and aHEI have been potentially associated with the risk of cardiovascular diseases and T2D, which can be used to assess the dietary pattern in a population with prediabetes. In addition, the FFQ has good reproducibility at estimating the dietary intake of the five-year period prior to the time of the dietary intake assessment [28,41].

## 5. Conclusions

In this nonurban population, subjects with prediabetes did not show different dietary patterns when compared with a normal glucose tolerance group. Women had a healthier dietary pattern in comparison with men. Physical activity may be a protective factor, helping to prevent the incidence of diabetes. This study supports the need to provide clinical strategies based on diet in combination with physical activity to prevent prediabetes and inhibit the development of type 2 diabetes. However, the discordance of the findings in different populations pointed to the need for further research in this field.

## Figures and Tables

**Table 1 nutrients-13-00252-t001:** Clinical characteristics of the study groups.

Characteristics	Normal Glucose Tolerance (*n* = 319)	Prediabetes (*n* = 216)	*p*
Age (years)	47.5 (12.8)	54.8 (12.2)	<0.001
Educational level			0.001
Non university	269 (85.1)	204 (94.4)	
Graduate or higher	47 (14.9)	12 (5.6)	
Tobacco exposure	164 (51.4)	118 (54.6)	0.520
Regular physical activity	240 (76.2)	134 (62.0)	0.001
BMI (Kg/m^2^)	25.3 (4.3)	27.5 (4.8)	<0.001
Waist (cm)	92.0 (11.9)	97.0 (12.4)	<0.001
SBP (mmHg)	119.0 (16.3)	126.0 (16.6)	<0.001
DBP (mmHg)	75.7 (10.1)	78.3 (9.9)	0.004
Hypertension	42 (13.2)	48 (22.2)	0.009
Dyslipidemia	51 (16.0)	71 (32.9)	<0.001
HbA1c (%)	5.3 (0.3)	5.8 (0.3)	<0.001
HbA1c (mmol/mol)	33.8 (2.8)	40.0 (3.1)	<0.001
Total cholesterol (mg/dL)	198.0 (38.1)	205.0 (33.1)	0.021
HDL-cholesterol (mg/dL)	58.5 (14.9)	58.9 (14.4)	0.760
LDL-cholesterol (mg/dL)	120.0 (31.1)	125.0 (30.0)	0.054
Triglycerides (mg/dL)	105.0 (91.6)	112.0 (64.3)	0.343

Data are shown as *n* (%) for categorical variables and mean (SD) for continuous variables. BMI, body mass index; DBP, diastolic blood pressure; HbA1c, glycated hemoglobin; HDL-cholesterol, high density lipoprotein-cholesterol; LDL-cholesterol, low density lipoprotein-cholesterol; SBP, systolic blood pressure. Tobacco exposure includes current and former smokers. *p* was calculated according to the method of Benjamini and Hochberg for multiple comparisons.

**Table 2 nutrients-13-00252-t002:** Dietary quality index of the study groups.

Variables	Normal Glucose Tolerance				Prediabetes				*p* Overall ^1^
	All (*n* = 319)	Men (*n* = 136)	Women (*n* = 183)	*p* Men vs. Women	All (*n* = 216)	Men (*n* = 88)	Women (*n* = 128)	*p* Men vs. Women	
aMED	3.2 (1.8)	3.0 (1.8)	3.3 (1.8)	0.140	3.4 (1.8)	2.9 (1.5)	3.7 (1.9)	0.001	0.164
aMED (tertiles)				0.163				0.018	0.189
T1 [0–3)	124 (38.9)	61 (44.9)	63 (34.4)		68 (31.4)	34 (38.6)	34 (26.6)		
T2 [3–5)	120 (37.6)	47 (34.6)	73 (39.9)		87 (40.3)	38 (43.2)	49 (38.3)		
T3 [5–8]	75 (23.5)	28 (20.6)	47 (25.7)		61 (28.2)	16 (18.2)	45 (35.2)		
aHEI	38.6 (7.3)	37.1 (7.9)	39.8 (6.6)	0.001	38.7 (6.7)	36.1 (6.2)	40.5 (6.9)	<0.001	0.877
aHEI (tertiles)				0.001				<0.001	0.431
T1 [20–36)	113 (35.4)	64 (47.1)	49 (26.8)		66 (30.6)	39 (44.3)	27 (21.1)		
T2 [36–43)	112 (35.1)	42 (30.9)	70 (38.3)		86 (38.9)	35 (39.8)	51 (39.8)		
T3 [43–64]	94 (29.5)	30 (22.1)	64 (35.0)		64 (29.6)	14 (15.9)	50 (39.1)		

Data are shown as *n* (%) for tertiles and as mean (SD) for continuous variables. aMED, alternate Mediterranean Diet score; aHEI, alternate Healthy Eating Index. *p* was calculated according to the method of Benjamini and Hochberg for multiple comparisons. ^1^
*p* overall for the comparation between normal glucose tolerance and prediabetes groups. T1, T2 and T3 are tertiles of aMED and aHEI, from T1 (poorer adherence) to T3 (better adherence). Square brackets indicate the inclusion of the value in the range and round brackets indicate the non-inclusion of the value in that range.

**Table 3 nutrients-13-00252-t003:** Multivariable analysis for the alternate Mediterranean Diet score (aMED) and the alternate Healthy Eating Index (aHEI) with the presence of prediabetes.

Variables	Crude OR (95% CI)	Adjusted OR (95% CI)	*p*	Crude OR (95% CI)	Adjusted OR (95% CI)	*p*
aMED moderate [3–5)	1.32 (0.88–1.98)	1.19 (0.75–1.87)	0.460			
aMED high [5–8]	1.48 (0.95–2.32)	1.26 (0.75–2.10)	0.379			
aHEI moderate [36–43)				1.31 (0.87–1.99)	1.32 (0.83–2.10)	0.246
aHEI high [43–64]				1.17 (0.75–1.81)	0.87 (0.52–1.46)	0.592
Age (years)	1.05 (1.03–1.06)	1.04 (1.02–1.05)	<0.001	1.05 (1.03–1.06)	1.04 (1.02–1.06)	<0.001
Sex (women)	1.08 (0.76–1.53)	1.14 (0.78–1.69)	0.503	1.08 (0.76–1.53)	1.19 (0.80–1.77)	0.403
Body mass index (Kg/m^2^)	1.11 (1.07–1.16)	1.09 (1.05–1.14)	<0.001	1.11 (1.07–1.16)	1.09 (1.05–1.14)	<0.001
Educational level, graduate or higher	0.34 (0.17–0.65)	0.51 (0.24–1.01)	0.061	0.34 (0.17–0.65)	0.54 (0.25–1.06)	0.083
Hypertension	1.88 (1.19–2.97)	0.87 (0.50–1.53)	0.638	1.88 (1.19–2.97)	0.90 (0.51–1.58)	0.720
Dyslipidemia	2.57 (1.70–3.89)	1.89 (1.19–3.00)	0.007	2.57 (1.70–3.89)	2.02 (1.27–3.22)	0.003
Physical activity	0.51 (0.35–0.75)	0.48 (0.31–0.72)	<0.001	0.51 (0.35–0.75)	0.49 (0.32–0.74)	0.001

aHEI, alternate Healthy Eating Index; aMED, alternate Mediterranean Diet score; BMI, body mass index.

## Data Availability

The data presented in this study are available on request from the corresponding author.

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
