# Peer review of "Mediterranean Diet and Healthy Eating in Subjects with Prediabetes from the Mollerussa Prospective Observational Cohort Study"

_nutrients, 2021, doi:10.3390/nu13010252_

Round 1

Reviewer 1 Report

Abstract 

Overall well written with important information given. However, p values without actual effect sizes do not mean much. Either provide the effect sizes or do away with the p values. 

Introduction

Succinct and provided sufficient background information on the topic. 

Methods

Section 2.3 - Do provide a summary of some of the measures eg. anthropometric if it has been explained in greater detail in another paper. Provide ethics approval number. 

Results

Similar to abstract, p values without effect sizes do not provide much meaning to the explanation. Consider highlighting very important findings in text with effect sizes and corresponding p values. 

Table 1 & 2 - The significance of analysing comparisons between gender is not clear eg, Table 1 - why are we looking for a difference in age between males and females with normal glucose tolerance and again in prediabetes groups? In my opinion, an overall comparison between normal glucose vs prediabetes groups will be sufficient. 

Section 3.2, Line 6 - Authors mentioned Table S3, Fig 1 & 2 which are not available. 

Table 3 - This is a very confusing table on multivariable logistic regression. Authors are advised to provide all crude OR analysis, followed by multivariable analysis. The statistical assumption made, covariates and specific type of multivariable logistic regression must be explained either here or in statistical analysis section. Were there any analysis of interaction, mediators or confounding factors done? 

Discussion

Sufficiently written.

Conclusion

Further elaboration on how the findings influence practice and recommendations for future research is required. 

Author Response

RESPONSE TO THE REVIEWER’S COMMENTS

Mediterranean diet and healthy eating in subjects with prediabetes from the Mollerussa prospective observational cohort study (manuscript reference nutrients-1058229).

We highly appreciate the input given by the Reviewers, which enabled us to greatly improve the quality of our Manuscript. We are hereby enclosing our point-by-point responses to each of the Reviewer’s comments. Please, note that in the revised version of the Manuscript, changes are marked with the track changes tool, so that they can be easily traced.

With regard to the request from the Editor to revise the text overlap, the Reviewer is kindly informed that this is mainly because of similarities with previous work of our group using this methodology or performing studies in the same study population. We have rewritten those portions of text in the Methods and Discussion sections. To facilitate the revision by the Editor and the Reviewers, apart from track changes in response to comments from the Reviewers, those paragraphs with edits to the text introduced on request from the Editor regarding the text overlap issue are marked with a yellow background.

REVIEWER 1

Abstract

Overall well written with important information given. However, p values without actual effect sizes do not mean much. Either provide the effect sizes or do away with the p values.

We thank the Reviewer for all his/her comments, and we agree with this comment. Therefore, following this Reviewer’s suggestion we have included the odds ratio and 95% CI with p values in the results of the abstract. Please, see changes with the track changes tool in the revised version of the manuscript, as follows:

Abstract

Lines 7-20: There was no difference in the mean aMED and aHEI scores between both groups (3.2 (1.8) in the normoglycemic group and 3.4 (1.8) in the prediabetes group, p=0.164 for the aMED, and 38.6 (7.3) in the normoglycemic group and 38.7 (6.7) in the prediabetes group, p=0.877 for the aHEI, respectively). Nevertheless, women had a higher mean of aMED and aHEI scores in the prediabetes group (3.7 (1.9), p=0.001 and 40.5 (6.9), p<0.001, respectively); moreover, they had a higher mean of aHEI in the group with normoglycemia (39.8 (6.6); p=0.001). No differences were observed in daily food intake between both study groups; consistent with this finding, we did not find major differences in nutrient intake between groups. In the multivariable analyses, the aMED and aHEI were not associated with prediabetes (OR: 1.19, 95% CI: 0.75-1.87; p=0.460 and OR: 1.32, 95% CI: 0.83-2.10; p=0.246, respectively); however, age (OR: 1.04, 95% CI: 1.02-1.05; p<0.001), dyslipidemia (OR: 2.02,  95% CI: 1.27-3.22; p=0.003) and BMI (OR: 1.09, 95% CI: 1.05-1.14; p<0.001) were positively associated with prediabetes. Physical activity was associated with a lower frequency of prediabetes (OR: 0.48, 95% CI: 0.31-0.72; p=0.001).

Introduction

Succinct and provided sufficient background information on the topic.

Methods

Section 2.3 - Do provide a summary of some of the measures eg. anthropometric if it has been explained in greater detail in another paper. Provide ethics approval number.

Following the Reviewer’s suggestion, we have included a statement describing the anthropometric measures collected in the study. In addition, the ethics approval number was already included at the end of this section. Please, see the revised version of the manuscript (track changes).

2.3. Clinical Data

Lines 3-4: Anthropometric measures i.e., body weight, height and waist, were determined using standard methods as previously described [26,27].   

Results

Similar to abstract, p values without effect sizes do not provide much meaning to the explanation. Consider highlighting very important findings in text with effect sizes and corresponding p values.

We agree with this Reviewer’s comment. Thus, we have included these data in the Results section. However, following the suggestion of the other Reviewer, we have removed the results regarding differences in clinical characteristics between men and women from Table 1. Therefore, the Reviewer may see clinical characteristics of both study groups in Table 1, and a description of these results in the first paragraph of the Results section.

All these changes can be seen by the Reviewer in the revised manuscript, as follows:

3. Results

 The clinical characteristics of the study groups are shown in Table 1. Subjects with prediabetes were older (mean (SD)=54.8 (12.2) and 47.5 (12.8); p<0.001, respectively), and had a lower educational level (94.4% and 85.1%; p=0.001, respectively), higher BMI (mean (SD)= 27.5 (4.8) and 25.3 (4.3); p<0.001, respectively), waist (mean (SD)=97.0 (12.4) and 92.0 (11.9); p<0.001, respectively), hypertension (22.2% and 13.2%; p=0.009, respectively), dyslipidemia (32.9% and 16.0%; p<0.001, respectively), HbA1c (mean (SD)=5.8 (0.3) and 5.3 (0.3); p<0.001, respectively) and total cholesterol (TC) (mean (SD)=205.0 (33.1) and 198.0 (38.1); p=0.021, respectively) compared with the normoglycemic group. In addition, they were more sedentary (62.0% and 76.2% with regular physical activity; p=0.001, respectively).

3.1. Dietary pattern and prediabetes

The dietary quality indexes of the study groups are shown in Table 2. In the group with prediabetes, the mean aMED and aHEI scores were higher in women compared with men (3.7 (1.9) and 2.9 (1.5), p=0.001 in the aMED, and 40.5 (6.9) and 36.1 (6.2) in the aHEI, p<0.001, respectively); moreover, a higher frequency of women in higher tertiles of aMED and aHEI was observed in comparison with men (35.2% and 18.2%; p=0.018 for the aMED, and 39.1% and 15.9%; p<0.001 for the aHEI, respectively). In addition, women with normal glucose tolerance also showed a higher aHEI compared with men (39.8 (6.6) and 37.1 (7.9); p=0.001, respectively). However, no differences were observed in the aMED and aHEI between both study groups; regarding aMED, the mean (SD) was 3.2 (1.8) in the group with normal glucose tolerance, and 3.4 (1.8) in the group with prediabetes (p=0.164); for aHEI, the mean (SD) was 38.6 (7.3) in the normoglycemic group, and 38.7 (6.7) in the prediabetes group (p=0.877). In terms of daily food consumption and nutrient intake, no differences were observed between both groups in any food or nutrient included in the aMED and aHEI (Table S1, Table S2). However, participants with normal glucose tolerance had a higher consumption of stearic acid in comparison with individuals with prediabetes (6.4 g/day and 5.9 g/day; p=0.011, respectively) (Table S2). 

Table 1 & 2 - The significance of analysing comparisons between gender is not clear eg, Table 1 - why are we looking for a difference in age between males and females with normal glucose tolerance and again in prediabetes groups? In my opinion, an overall comparison between normal glucose vs prediabetes groups will be sufficient.

We agree with this Reviewer’s comment. Therefore, we have deleted the stratification of clinical characteristics by gender. Please, see the modified table in the revised version of the manuscript.

Section 3.2, Line 6 - Authors mentioned Table S3, Fig 1 & 2 which are not available.

We apologize for this problem. Actually, this was an error as we kept the mention to Table S3, Fig 1 & 2 of a draft version of the paper that was not the version finally submitted to the journal. Actually, these results were all included as Table 3 in the final version of the paper. This has now been amended and the reader is referred to Table 3.

Table 3 - This is a very confusing table on multivariable logistic regression. Authors are advised to provide all crude OR analysis, followed by multivariable analysis. The statistical assumption made, covariates and specific type of multivariable logistic regression must be explained either here or in statistical analysis section. Were there any analysis of interaction, mediators or confounding factors done?

We appreciate the Reviewer’s suggestion. As the Reviewer can see, we have provided crude and adjusted OR in Table 3 of the revised manuscript. The analyses were conducted to examine the association between the dietary pattern and prediabetes. Moreover, according to the background, the multivariable analyses included clinical variables related with prediabetes and the dietary pattern as potential confounding factors to support the hypothesis of the study using the enter method. Therefore, we have included a short text explaining that the Enter method was applied with the multivariable analysis in the Methods section, as follows:

2.4. Statistical analysis

Lines 13-14: […] with diabetes using the Enter method to support the hypothesis of the study.

Furthermore, we have just checked the interactions sex-aHEI and sex-aMED as these were plausible interactions, and we did not find statistical significance (OR: 1.01, 95% CI: 0.96-1.52, p=0.612 for the interaction sex-aHEI, and OR: 1.18, 95% CI: 0.95-1.48, p=0.133 for the interaction sex-aMED). Therefore, we have included a sentence in the Results section explaining this. Changes are included in the revised version of the manuscript, as follows:

3.2. Clinical factors and prediabetes

Lines 8-9: In addition, we could not find an interaction between sex and aHEI or sex and aMED (no statistical significance in the multivariable models; data not shown).

Discussion

Sufficiently written.

Conclusion

Further elaboration on how the findings influence practice and recommendations for future research is required.

We thank the Reviewer for this comment. Therefore, we have included the following statement in the Conclusions section of the revised manuscript.

5. Conclusions

Lines 4-7: This study supports the need to provide clinical strategies based on diet in combination with physical activity to prevent prediabetes and further the development of type 2 diabetes.

Reviewer 2 Report

This is a very interesting study aiming 1) to assess the relationship between dietary patterns and the prevalence of prediabetes, and 2) to investigate factors associated with prediabetes. Analysis have been performed in 535 men and women, 216 individuals with prediabetes and 319 individuals with a normal glucose tolerance, enrolled in the Mollerussa prospective cohort study. The dietary pattern was assessed using the alternate Mediterranean diet score (aMED) and the alternate Healthy Eating Index (aHEI). Authors did not find any differences between the two groups in terms of dietary patterns, as well as in terms of daily food consumption. On the contrary, age, the presence of dyslipidemia, and BMI were associated with prediabetes condition. 

The study is well written and the methodology well described. However, I have some comments to add:

  1. Please clarify in the methods (study design) if people following a special diet (i.e., for prediabetes condition, or others conditions) were excluded from the analysis. This is a very important point because the inclusion of people following a specific diet does not permit to define the energy and nutrient composition of the diet, as well as the food consumption, as "habitual diet". Therefore, this is an important limitation to add also in the discussion.
  2. Authors did not find any differences in the two dietary patterns analyzed. Have you performed a specific analysis on each food group included in the aMED and aHEI. A possible difference could appair.
  3. Please add one table on the energy and nutrient composition of the diet. Please report proteins, carbohydrates, sugar, fat,saturated fat, etc., in terms of percentage of total energy.
  4. Supplemental table 2: Please check the p-value. It is very strange that in more cases is 0.986
  5. Please add in the discussion a sentence on the importance of factors as age, bmi, etc., on the prevalence of prediabetes more than dietary habits. 

Author Response

RESPONSE TO THE REVIEWER’S COMMENTS

 Mediterranean diet and healthy eating in subjects with prediabetes from the Mollerussa prospective observational cohort study (manuscript reference nutrients-1058229).

We highly appreciate the input given by the Reviewers, which enabled us to greatly improve the quality of our Manuscript. We are hereby enclosing our point-by-point responses to each of the Reviewer’s comments. Please, note that in the revised version of the Manuscript, changes are marked with the track changes tool, so that they can be easily traced.

With regard to the request from the Editor to revise the text overlap, the Reviewer is kindly informed that this is mainly because of similarities with previous work of our group using this methodology or performing studies in the same study population. We have rewritten those portions of text in the Methods and Discussion sections. To facilitate the revision by the Editor and the Reviewers, apart from track changes in response to comments from the Reviewers, those paragraphs with edits to the text introduced on request from the Editor regarding the text overlap issue are marked with a yellow background.

REVIEWER 2

This is a very interesting study aiming 1) to assess the relationship between dietary patterns and the prevalence of prediabetes, and 2) to investigate factors associated with prediabetes. Analysis have been performed in 535 men and women, 216 individuals with prediabetes and 319 individuals with a normal glucose tolerance, enrolled in the Mollerussa prospective cohort study. The dietary pattern was assessed using the alternate Mediterranean diet score (aMED) and the alternate Healthy Eating Index (aHEI). Authors did not find any differences between the two groups in terms of dietary patterns, as well as in terms of daily food consumption. On the contrary, age, the presence of dyslipidemia, and BMI were associated with prediabetes condition. 

The study is well written and the methodology well described. However, I have some comments to add:

  1. Please clarify in the methods (study design) if people following a special diet (i.e., for prediabetes condition, or others conditions) were excluded from the analysis. This is a very important point because the inclusion of people following a specific diet does not permit to define the energy and nutrient composition of the diet, as well as the food consumption, as "habitual diet". Therefore, this is an important limitation to add also in the discussion.

We thank the Reviewer for his/her comments. In this study, we excluded those participants who were prescribed a therapeutic diet or had any condition that required specific medical nutrition therapy. We agree with the Reviewer’s comment, and therefore, we have included this exclusion criteria in the Methods section. Please, the Reviewer can see the changes in the revised version of the manuscript (track changes).

3.1. Study design

Lines 13-14: […] or major psychiatric disorders, and the presence of any condition requiring any diet treatment [26,27].

  1. Authors did not find any differences in the two dietary patterns analyzed. Have you performed a specific analysis on each food group included in the aMED and aHEI. A possible difference could appair.

We appreciate the Reviewer’s comment. We already analyzed the differences between daily food consumption and nutrient intake between both groups, and we only found a higher consumption of stearic acid in participants with normal glucose tolerance compared with the prediabetes group. These results are described in the Results (section, 3.1. Dietary pattern and prediabetes) of the manuscript. The Reviewer can find this information in Table S1 and Table S2. The daily food consumption and nutrient intake included food groups and nutrients used to calculate the aMED and aHEI scores. Therefore, we have included a statement in the manuscript detailing that we did not find any difference between food groups and nutrients considered in the aMED and aHEI, as follows:

3.1. Dietary pattern and prediabetes

Lines 12-14: In terms of daily food consumption and nutrient intake, no differences were observed between both groups in any food or nutrient included in the aMED and aHEI (Table S1, Table S2).

  1. Please add one table on the energy and nutrient composition of the diet. Please report proteins, carbohydrates, sugar, fat,saturated fat, etc., in terms of percentage of total energy.

 We appreciate the Reviewer’s comment. Therefore, following this suggestion, we have included percentages of total energy intake of carbohydrates, sugar, proteins, total fat, saturated fat, polyunsaturated fat and monounsaturated fat in Table S2. Please, see changes in the revised version of the Supplemental Files.

  1. Supplemental table 2: Please check the p-value. It is very strange that in more cases is 0.986

 We have checked these figures and can confirm that values are correct. This is explained by an effect of the high number of subvariables evaluated, and the correction used, that produced a rise in the p-values and a trend approaching to 1.  Please, note that we used the correction proposed by Benjamini & Hochberg (1995) ("BH" or its alias false discovery rate) with the "p.adjust" function from the R Stats package (version 3.6.3). This analytical methodology yielded a threshold p value for a number of variables (i.e. 0.986). 

  1. Please add in the discussion a sentence on the importance of factors as age, bmi, etc., on the prevalence of prediabetes more than dietary habits. 

Following this Reviewer’s suggestion, we have included a statement in the Discussion section indicating the important role of these factors in the development of prediabetes and T2D. Please, see changes in the revised manuscript.

4. Discussion

Lines 67-69: Moreover, clinical factors such as age, BMI and dyslipidemia have a major role in the development of prediabetes and T2D; therefore, this should be further considered in future research.

Round 2

Reviewer 1 Report

I am satisfied with the changes made by the authors and have no further comments to make.